# OpenReview forum: "Building Socio-culturally Inclusive Stereotype Resources with Community Engagement"
_NeurIPS.cc/2023/Track/Datasets_and_Benchmarks — NeurIPS 2023 Datasets and Benchmarks Poster_

### Official Review · Reviewer_9GE9 · 2023-07-20
**A small but valuable dataset**

**Rating:** 7
**Confidence:** 4
**Correctness:** The analyses look correct.
**Clarity:** Very clear.

**Strengths:**

The paper is very well structured and presented. Despite some language issues pointed out in the opportunities for improvement section, the paper is, in general, very readable, understandable, and well-explained.

The collection of the dataset is well-motivated and exciting potential applications are touched upon, as well.

The resulting dataset is likely going to be valuable to the community, and the current wide adoption of LLM solutions and applications is making it even more relevant.

The analyses, despite not being too extensive, reveal valuable insights, such as that offensiveness scores range across identities; and that people from different regions remember and report different stereotypes, often about their own identity and sometimes about others.

**Additional Feedback:**

%

**Documentation:**

Well documented for the most part.

**Ethics:**

Well discussed.

**Limitations:**

One potential limitation is that all the respondents are students. The authors aimed at diversyfying them at other characteristics and that is excellent, but the narrow age range and its uniformity should be mentioned under limitations.

**Opportunities For Improvement:**

Despite the number of stereotypes is increased significantly compared to existing similar datasets (i.e., from ~200 to ~1200), it is not clear whether the new stereotypes add so much value. For instance, the authors could provide a bit more insight into the type of novel stereotypes; how they differ from what was used before (e.g., by listing the most popular new ones in Supplementary Information or by providing a topic analysis on them).

The authors could present some additional results beyond exemplary for all their analyses. For instance, could the offensiveness scores be visualised in some way for all the stereotypes collected? What is the overall average, and which are the top and least offensive stereotypes/identities?

Similarly, for Wikipedia analysis, could you show statistics on the average and standard deviation numbers of articles associated with all stereotype pairs?


There are several typos (e.g., free from instead of free form, identity) and grammatical issues in the sections Increased Number and Diversity of Identities in Stereotypes, and State, ethnic, and regional stereotypes.

There are formatting issues in the E.g. Tuples Entailed (e.g., missing comma or extra semicolon) in Table 5.

**Relation To Prior Work:**

Well described and positioned.

**Summary And Contributions:**


The paper presents a dataset crowdsourced in a carefully designed way to include mentions of stereotypes (in the form of pairs: identity -- attribute) across India. The stereotypes collected cover 4 main regions in India, and a large number of urban and suburban areas within, from more than 500 participants. The participants were solicited among students, with the idea to cover various social groups among them, as the universities have the rule to accept students from various backgrounds. In this work, it is also shown that the collected dataset uncovers 100s of more identities and attributes than that are present in existing similar stereotype datasets. Moreover, an NLI analysis using various embeddings shows that there are different levels of offensiveness associated with attributes of different identities (e.g., the least levels with Hindu and the most with Muslim).

---

> ### Author Response · Authors · 2023-08-16
>
> Thank you for the valuable feedback and comments.
>
> On value of stereotypes collected:
> We observe that the increase in number and diversity of salient, local stereotypes in English add immense value in implementing safeguards in English language models, as seen in Section 6 (especially in Tables 4, and 5). In table 5 specifically, we look at how caste and religion based stereotypes that are not present in other resources but are collected in SPICE show up in models (for e.g., RoBERTa stereotypes in 31.9% of test samples wrt caste and 24.4% of test samples wrt religion).
> As English language models are prevalent in India, it is important that such evaluations incorporate resources that are cognizant of local stereotypes and harms.
> To help highlight this importance in inclusion of this resource in evaluation systems, we have now added a summary paragraph at the end of Section 6.2 which discusses the major takeaways.
>
> On offensiveness calculations and results:
> Thank you for the questions on offensiveness calculations. We reflected on this and added details and a visualization in the revised submission as per the recommendation.
> Specifically, we have now added further details on how offensiveness of attributes was determined in Section 6.3 using a pre-existing resource we collected and published (https://github.com/google-research-datasets/seegull; Jha et al 2023). Further, Section 6.3 discusses some of the most stereotypical attributes in the dataset, and also some of the identities with disparate offensive stereotypes. However, it also notes that this may not be truly reflective of what the most offensive attributes are because there is only a small subset (217) of attributes from SPICE that were present in the resource used for determining offensiveness. Further, offensiveness of an attribute is subjective, and estimation of how offensive something is needs to be conducted locally. With these caveats, we have now added a visualization in Section 6.3 of what some of the identities with most of least offensive attributes are.
>
> On formatting, spelling, and grammar:
> Thank you so much for pointing out the mistypes and issues in text. We have now done a deep formatting, spelling, and grammatical check in the revised version submitted.

---

> > ### Comment · Reviewer_9GE9 · 2023-08-29
> >
> > Dear authors,
> > Thanks for addressing the comments. Hence, I will keep my rating as is.

---

### Official Review · Reviewer_PwCG · 2023-07-24
**Useful dataset containing stereotypes on intersectional groups in India**

**Rating:** 6
**Confidence:** 4
**Correctness:** Yes, theory, results, and analysis se…
**Clarity:** Yes, very clear.

**Strengths:**

+ Authors present the most comprehensive list of stereotypes across intersectional categories until today (to the best of my knowledge).

+ They make this list available for further use, and they show how it can be used to uncover biases in NLP models.

+ Authors take a situated model design approach and ask individuals explicitly about stereotypes regarding their identity.



**Additional Feedback:**

-

**Documentation:**

Yes, authors disclose survey forms and the dataset.

**Ethics:**

No ethical concerns, authors have a dedicated section and also took data subject's privacy into consideration.

**Limitations:**

- Authors do mention limitations in the universality of collected stereotypes, which is much appreciated. It would also be good to explain limitations in regard to age.

- Authors could also perform additional experiments to showcase the value of the dataset, besides doing a single experiment with LLMs. In this way, they can show to machine learning practitioners additional way that this list is useful.



**Opportunities For Improvement:**

- Authors collect stereotypes only from university students to ensure that they find adequate individuals across intersectional categories. Nonetheless, this introduces an age bias in the dataset, filtering the experiences of older people and the stereotyping they faced.

- Although I support the decision to ask impacted populations about what stereotypes they face, stereotyping is generally in the "eye of the beholder." An integrative perspective should ideally also include the stereotypes individuals have about other identities, apart of their own. For example, I can only recall stereotypes other people have about me, which I became conscious of (and are usually also of more negative valence). This puts a limitation on what can be discovered by the chosen survey methodology.

- Why doesn't table 5 also have a comparison with/without Spice, as table 4?


**Relation To Prior Work:**

Yes, the list of stereotypes is much more extensive that existing ones.


**Summary And Contributions:**

Authors present an extensive list of stereotypes across various intersectional categories in India. They create this list by surveying university students in various areas of India. By adopting an open-ended survey study design, they are able to collect much more attributes than prior attempts. Authors showcase the utility of the list by showing how it can be used to uncover additional biases of LLMs. They also make the dataset available to be used by the research community.

---

> ### Author Response · Authors · 2023-08-16
>
> Thank you for the valuable feedback and comments.
>
> On the limited age representation in the data:
> We completely agree. We indeed do have limitations pertaining to views only being representative of certain age groups. Our aim in this work was to diversify representation wrt gender, socio economic status, region, language, religion, and caste, which is why we selected public aided or run universities / colleges. However, it limits representation on other axes of identities such as age, languages known, access to education, etc which we leave to future work. We now highlight this design choice in Section 4.2 and add how it limits the work in Section 7.
>
> On what stereotypes a respondent is expected to contribute:
> Thank you for the feedback. In this study, we don't distinguish (nor ask for) whether the respondent believes in the provided stereotypes or not, as our intention is not to study the participants, rather to build a more comprehensive repository of possible stereotypes present in Indian society, so as to equip model evaluations better. We have added this clarification in Section 4.1 of our revised submission.
>
> On experiments:
> For stereotypes related to regions in India in Table 4, there are existing datasets to compare against. However, to the best of our knowledge, no other similar stereotype repositories or lists exist for the identities along caste and religion for comparison in Table 5, and hence we only show results with SPICE.
>
> We completely agree on demonstrating the need of such resources in evaluations of models. In Section 6 we elucidate some of the different ways in which SPICE can be used to test data and models for stereotyping harms. In particular, in Section 6.2, we see how using SPICE, significant amounts of stereotyping --along caste, region, and religion-- is observed across different language models. This are illustrations of how to use this dataset effectively and many more such evaluation paradigms can be created, which is beyond the scope (and space available) of the paper. We sincerely hope that the data resource we create here gets used in many such ways for better evaluations.

---

> > ### Comment · Reviewer_PwCG · 2023-08-30
> >
> > Dear authors, thanks for the clarifications and improvements. Based on them, I am keeping my score as is.

---

### Official Review · Reviewer_9hoA · 2023-07-24
**Community stereotype survey and dataset**

**Rating:** 6
**Confidence:** 3
**Correctness:** I do not have any correctness concerns.

**Strengths:**

I really appreciate the authors' motivations and their methodology. Focusing on non-western stereotypes is well-motivated, and collecting community responses seems like the best course of action for collecting this data.

The paper itself is generally well-structured, clear and easy to follow.

**Additional Feedback:**

None

**Clarity:**

The paper is well-written with the exception of a couple of points highlighted above that were not clear to me.

**Documentation:**

No complaints besides the clarifications requested above.

**Ethics:**

I only have one significant concern here -- due to the sensitive nature of the data -- can it, and is it likely to be misused, purposefully or accidentally?

**Limitations:**

I think overall the authors made very reasonable choices given the challenging task. I am not sure how representative are the answers as the age sample is rather limited and these are University students, but I can't think of a better way to conduct this that would be practical.

I think the authors do a good job of highlighting the limitations of their work, but the impact of these could be further discussed.

I also don't know how much they experimented with the survey design, but I would be interested to know if they did test several options.

**Opportunities For Improvement:**

Mostly I was left with several questions and things I wished the authors will discuss:

1. I was not clear whether respondents were asked about stereotypes they *believed* in or that they *heard* of. Both can have value, but they are quite different things. I would have liked to see this clarified and discussed if possible.

2. It strikes me that the level of offensiveness of the stereotype is rather critical to this resource. However, I was unsure how this was determined and by whom.

3. I would have liked to see a section on how can this resource be used? (for good)

4. I would have liked to read more about the potential impact of the sampling choice.

5. Explain more regarding the decision to create this dataset in English

**Relation To Prior Work:**

I was unclear if there are any existing datasets that cover India-specific stereotypes.

Related to the above, maybe more could be mentioned in terms of how these should or could be handled (but I do appreciate space is tight!)

**Summary And Contributions:**

By surveying university students, the authors create a dataset of 'stereotypes' focused on India. This newly created resource is novel in its method of creation and produces a much larger list of stereotypes.

---

> ### Author Response · Authors · 2023-08-16
>
> Thank you for the valuable feedback and comments.
>
> We first answer the numbered questions posed in the ‘Opportunities For Improvement’ section:
> 1. We don't distinguish (nor ask for) whether the respondent believes in the provided stereotypes or not, as our intention is not to study the participants, rather to build a more comprehensive repository of possible stereotypes present in Indian society, so as to equip model evaluations better. We clarify this further in Section 4.1 of our revised submission.
>
> 2. The level of offensiveness was determined using a pre-existing resource we collected and published (https://github.com/google-research-datasets/seegull). Please see the paper Jha et al. (2023) for more details about the collection of these offensiveness ratings. In order to address your feedback, we have added a brief summary of the details on the collection and significance of the values of offensiveness in Section 6.3.
>
> 3. We agree on the importance of demonstrating utility of such a dataset. In fact, Section 6.2 already describes the utility of SPICE in an evaluation context of the natural language inference task, which when reflective of actual locally situated stereotypes, can determine potential harmful effects of models more accurately. This can be seen in Table 4, where with SPICE substantially more regional stereotypes were detected in language models (for e.g., 5.6% in ELECTRA, 8.5% in XLNET). Further, in Table 5 we see how stereotypical associations for caste and religion are seen across all models investigated (for e.g., in RoBERTA caste associations 31.9%, religion associations 24.4%), which were earlier undetected.
> We have now added a summary paragraph at the end of Section 6.2 highlighting this contribution and its importance in evaluations.
>
> 4. We chose to distribute surveys in government run or aided universities/colleges in India for collection of this stereotype data as an attempt to have a diverse respondent pool. However, this choice too can be limiting and result in a respondent pool that is not representative of other identities, which we discuss in Section 4.2 on Survey Distribution and in Section 7 under Limitations. Several such slices of diverse pools need to be engaged with for analysis on the impact of sampling of respondent pools, which was beyond the scope of this work, but would be extremely valuable to investigate deeply in future work. In this paper, we aim to lay the foundation on which such work can be built.
>
> 5. We choose to conduct this study in English since language models currently deployed in India are predominantly in the English language, but stereotype evaluations do not consider salient stereotypes in India. Safeguards in the English language are thus urgent to be developed.
> Further, this work is intended as ground work gaining us proof-of-concept of collecting such a resource at the grass root level using free form text based surveys. While challenging to complete in one language only, the success of extracting a resource this way will act as the foundation towards our future work where we shift focus to the multitude of languages used in India.
> Thank you for this very significant question. We have added these details to clarify to all readers our motivation in Section 4.1.
>
> On survey design experimentation:
> Thank you for the question. In the study, we experimented with the mode of discussing with participants to inform what stereotypes are. The videos in the Appendix were deemed most explanatory and easy to interpret the survey with. To address this question, we have now added this iterative process in the paper in Section 4.1.
>
> On potential misuse of the data:
> Thank you so much for pointing this out. It indeed is possible that the work and data collected could be either intentionally or inadvertently misused. Keeping this in mind, we already included the specific intended usage of model evaluation specified in our Data Card. Thanks to your recommendation, we have now added the same caution against misuse in an additional paragraph in the 'Limitations and‘Broader Impacts’ segment in Section 7 of our revised submission.

---

> > ### Comment · Reviewer_9hoA · 2023-08-31
> >
> > Dear authors, thank you for these clarifications! I am happy with these explanations and will be keeping my score the same.

---

### Official Review · Reviewer_eMmS · 2023-07-25
**Nice and needed effort!**

**Rating:** 6
**Confidence:** 3
**Correctness:** The evaluations on the dataset seem t…

**Strengths:**

I believe that the methodology proposed in the paper is an important step forward in the field of fairness. As the authors claim, it is unfortunately common to consider societal biases only from a western-society perspective, without taking into account that fairness is manifested in different ways in other societal context. This paper is a good example of how to take diversity into account.

**Additional Feedback:**

I am happy to participate to the open discussion. However, I will have no internet connection from the 2nd of August until the 15th. I will come back to work on the 16th and I'll be able to reply to any of your questions/comments/revisions only after that date.

**Clarity:**

The paper is well-written. Maybe the abstract and the introduction, however, create some sort of confusion. I would suggest that the authors try to simplify some of the terms utilized in those two paragraphs and try to be more direct in stating the motivations and intentions of the paper.

**Documentation:**

The documentation is exhaustive with the only issue of some ethical questions that could maybe be addressed more explicitly (see "Ethics" section in the review).

**Ethics:**

The dataset seems to be collected in an ethical way, and the authors also provide the informed consent shown to the participants. However, given the nature of this study, I believe it would be beneficial to add also some more explicit information about the stored metadata (especially when it comes to sensitive data). Other authors have been using "datasheet for datasets" but I believe Neurips also suggests other good resources that would make the ethical understanding of the paper easier.

**Limitations:**

The authors properly address the limitations of their study and emphasize the intended use of their dataset.

**Opportunities For Improvement:**

Maybe the name of the dataset could be in the title? Also, maybe it would be beneficial if the title explicitly said that the analyzed context is the Indian society.

In the revised version of the paper, I would suggest an additional paragraph that summarizes the main outcomes from the experiments, while also suggesting other possible directions of research that this dataset opens.

**Relation To Prior Work:**

The "Related Work" session is well-written and particularly clear. Dividing it in two different paragraphs, it states why this contribution is important for the overall field of fairness, and it justified the used methodologies properly.

**Summary And Contributions:**

The authors build resources to evaluate stereotypes in language models for the specific case of the Indian society. To do so, they develop a community-engaged methodology.

---

> ### Author Response · Authors · 2023-08-16
>
> Thank you for the valuable feedback and comments.
>
> On summarizing main takeaways from experiments:
> Thank you very much for this suggestion to make the takeaway from the paper better! We have added an additional paragraph in Section 6.2, synthesizing the overall observations of improvement in evaluations using SPICE, and also recommending other ways to potentially improve upon such measurements of language models.
>
> On suggested updates to the title of the paper:
> Thank you for the feedback on the title and how it could benefit from specificity. Since we will not be able to update the title until camera-ready submission, we will reflect deeply on your recommendation further and consider adding it in the intervening time.
>
>
> More details about the data:
> We completely agree with the need to be transparent about the metadata and other details about the dataset collection and intended uses. In fact, we already included a Data Card (Pushkarna et al. 2022; https://dl.acm.org/doi/fullHtml/10.1145/3531146.3533231) for SPICE dataset in the supplementary information folder. Data card is a more recently proposed transparency artifact, that is similar in purpose to datasheet.

---

> > ### Comment · Reviewer_eMmS · 2023-08-26
> > **Thank you**
> >
> > Dear authors, thank you for your answer. For the time being, I will maintain my evaluation on 6, in agreement with what other reviewers have stated.

---

### Author Response · Authors · 2023-08-16

We thank all the reviewers deeply for the very thoughtful and detailed reviews, feedback, and comments.
We have reflected upon all the questions and suggestions at length and have revised and submitted our paper accordingly. In this revised version that you see, the newly added and/or edited text will appear in blue to help identify the changes we made as per your recommendations and very valuable feedback. We also point to these changes in the paper in our answers to your individual questions and reviews.

We have also replied to each of your specific questions, comments, and feedback overall individually in our comments below. We really appreciate the very constructive feedback we got and have tried our best to address it both in our responses and the revised paper. Please do let us know if there are any further clarifications we can make, or if there are any additional or follow up comments or questions.

---

### Decision · Program_Chairs · 2023-09-22

**Decision:**

Accept (Poster)

**Comment:**

The authors make a good effort to collect stereotypes that are larger than the previous datasets. I feel that this will promote further research in this area in future. However there a few points that I want the authors to consider in the final version:

- Many of the stereotypes are just inexpressible in English. For instance in Bengal there are stereotypes associated with two groups -- ঘটি and বাঙাল. These stereotypes and their context are impossible to express in English. The authors should acknowledge this limitation. Further it is not fully true that there are no models that are suitable for Indian languages; for example MuRIL, IndicBERT, ABUSE-XLMR etc.

- How would the results be affected if the more fashionable decoder only LLMs are used: gpt-3.5, gpt-4, BARD etc.

- Perception of stereotypes actually change with education. I would have wanted to see the results of their stereotype analysis based on the education level of the participant.